# Recurrent Implication of Striatal Cholinergic Interneurons in a Range of Neurodevelopmental, Neurodegenerative, and Neuropsychiatric Disorders

**DOI:** 10.3390/cells10040907

**Published:** 2021-04-15

**Authors:** Lauren A. Poppi, Khue Tu Ho-Nguyen, Anna Shi, Cynthia T. Daut, Max A. Tischfield

**Affiliations:** 1Human Genetics Institute of New Jersey, Rutgers, The State University of New Jersey, Piscataway, NJ 08854, USA; lp638@hginj.rutgers.edu; 2Child Health Institute of New Jersey, Robert Wood Johnson Medical School, New Brunswick, NJ 08901, USA; kt.hn@rutgers.edu (K.T.H.-N.); as2800@rutgers.edu (A.S.); ctd50@scarletmail.rutgers.edu (C.T.D.); 3Tourette International Collaborative (TIC) Genetics Study, Rutgers, The State University of New Jersey, Piscataway, NJ 08854, USA; 4Department of Cell Biology and Neuroscience, Rutgers, The State University of New Jersey, Piscataway, NJ 08854, USA

**Keywords:** striatum, interneuron, cholinergic, neuropsychiatric, movement disorder, neurodevelopment

## Abstract

Cholinergic interneurons are “gatekeepers” for striatal circuitry and play pivotal roles in attention, goal-directed actions, habit formation, and behavioral flexibility. Accordingly, perturbations to striatal cholinergic interneurons have been associated with many neurodevelopmental, neurodegenerative, and neuropsychiatric disorders. The role of acetylcholine in many of these disorders is well known, but the use of drugs targeting cholinergic systems fell out of favor due to adverse side effects and the introduction of other broadly acting compounds. However, in response to recent findings, re-examining the mechanisms of cholinergic interneuron dysfunction may reveal key insights into underlying pathogeneses. Here, we provide an update on striatal cholinergic interneuron function, connectivity, and their putative involvement in several disorders. In doing so, we aim to spotlight recurring physiological themes, circuits, and mechanisms that can be investigated in future studies using new tools and approaches.

## 1. Introduction

It has been known since the time of Ramón y Cajal (1911) that the corpus striatum contains many diverse neuronal subtypes. The vast majority (~95%) of the striatal cell population are gamma aminobutyric acid (GABA) containing medium spiny neurons (MSNs) [1]. The remaining ~5% are regulatory interneurons [2]. What functional influence could such a small number of cells really have? While they may be sparse, striatal interneurons exert a considerable influence over the activity of local circuits and are essential components of the striatal machinery [3,4,5,6,7]. Broadly, there are two major types of striatal interneuron: cholinergic interneurons (ChINs) and GABAergic interneurons. Striatal GABAergic interneurons can be further divided into distinct subpopulations that are being constantly recategorized as our ability to profile these neurons on a molecular and functional level increases. These include parvalbumin-expressing (PVINs), neuropeptide Y-expressing neurogliaform (NPY-NGFINs), somatostatin-expressing (SSTINs), calretinin-expressing, and tyrosine hydroxylase-expressing (THINs) interneurons [1,5]. Each subtype is distinguished by unique gene expression profiles that endow specialized morphological features, connectivity patterns, electrophysiological properties, and roles in striatal processing.

Two of the most highly studied populations of striatal interneurons are ChINs and PVINs, which are often regarded as analogous to two groups of neurons frequently described in in vivo physiological studies—tonically active neurons (TANs) and fast-spiking interneurons (FSIs), respectively [8,9,10]. ChINs constitute ~1% of the total striatal cell population in rodents and humans [2,11,12], but their large axonal arbors enable acetylcholine (ACh) release across vast striatal territories [13]. As a result, the striatum is one of the richest sources of ACh in the brain [13,14,15], often illustrated by intense acetylcholinesterase (AChE) immunoreactivity [16]. ACh has broad influences across many cell types, both pre- and post-synaptically, because of the ubiquitous expression of ACh receptors in the striatum [17]. ChINs are aptly described as “gatekeepers” of striatal circuitry and are frequently implicated in neurodevelopmental, neuropsychiatric, and neurodegenerative disorders [18,19].

Here, we review the contribution of ChINs to both normal and disordered physiological processes in the brain, starting with their development, intrinsic physiology, and functional integration into circuits, before discussing recent findings in models of several different disorders. There are several useful recent reviews in this area [20,21,22,23,24].

## 2. Development of Striatal Cholinergic Interneurons

The striatum is an integrative mix of molecularly, functionally, and anatomically diverse neuronal populations. Both MSNs and striatal interneurons arise from distinct proliferative zones in the telencephalon [25]. The homeobox transcription factor *Nkx2.1* is highly expressed in interneuron progenitors that tangentially migrate away from these proliferative zones and is necessary to partition these cells into separate streams bound for the cortex or striatum [26]. As they migrate, cortical-bound GABAergic interneurons rapidly downregulate *Nkx2.1* expression, while upregulating expression of semaphorin receptor(s) neuropilin1 and/or neuropilin2 [27,28]. Semaphorin signaling acts to repel cortical interneurons away from the *Sema3a* and *Sema3f*-positive striatum and towards the cortex [29]. By contrast, striatum-bound interneurons maintain *Nkx2.1* expression, which functions to downregulate the expression of neuropilins [30].

Striatal ChINs are among the earliest-born striatal interneurons [31,32], generated between embryonic day 11.5 (E11.5) and E14.5 in mice, and E12 and E17 in rats [27,31,32,33]. ChINs originate from the preoptic area, the septum [27,34,35], and the medial ganglionic eminence (MGE), the latter of which is a major proliferative zone for both cholinergic and GABAergic interneurons [36,37]. Approximately 50% of striatal ChINs co-express *Nkx2.1* and *Zic4*, indicating that they are of septal origin, and the majority of ChINs maintain *Nkx2.1* expression into maturity [35,38]. Conditional loss of *Nkx2.1* in the mouse septum results in fewer ChINs in the rostral striatum and impaired performance in a T-maze reward task [35]. It has been suggested that loss of *Nkx2.1* both pre- and postnatally leads to cell death as opposed to fate-switching, indicating its importance across different developmental phases [28]. *Gbx2* is another homeobox transcription factor that is highly expressed in tangentially migrating cells derived from the mouse MGE that give rise to striatal ChINs [33]. *Gbx2* is required for ChIN development as loss of *Gbx2* results in fewer ChINs, abnormal distribution across the striatum, and changes to neurite complexity [33].

Although overlapping transcription factor expression is noted across different striatal interneuron subtypes, commitment to a cholinergic or a GABAergic cell fate seems to be decided by competing expression of LIM homeodomain members *Lhx6* or *Lhx8* (also known as *L3*/*Lhx7*), both of which require *Nkx2.1* expression [27]. Neurons that highly express *Lhx6* are likely to commit to a GABAergic fate, while *Lhx8*+ neurons can become either GABAergic interneurons or ChINs [39,40]. *Lhx8* is also thought to be upstream from *Gbx2* since *Lhx8* knockout reduces *Gbx2* expression [41]. Another downstream target of *Lhx8* is LIM homeodomain-transcription factor islet-1 (*Isl1*) [38]. *Isl1* represses *Lhx6* expression and is thought to promote cholinergic cell fate by forming a hexamer complex with *Lhx8* that can bind cholinergic-specific enhancers to promote the expression of cholinergic pathway genes [42]. Conditional inactivation of *Lhx8* from committed ChINs causes these neurons to lose choline acetyltransferase (ChAT)-specific properties and instead acquire molecular and morphological features characteristic of GABAergic interneurons [43]. Thus, sustained expression of *Lhx8* is required to maintain cholinergic interneuron identity in postmitotic neurons.

Striatal ChINs are not a homogeneous population. A recent study reported that 50% of striatal ChINs co-express *ChAT*, *GAD65*, *Lhx6* and *Lhx8*, and co-release ACh and GABA [44]. Thus, ChINs display considerable molecular and functional heterogeneity that is likely imparted by differing developmental pathways [24,35,44]. A recent pre-print from Ranjbar-Slamloo et al. suggests that the transcription factor *Er81* is required for the proper anatomical and functional maturation of ChINs [45]. Going forward, it will be useful to identify additional transcription factors to help delineate different subpopulations of ChINs [46]. The integration of ChINs into striatal circuits likely requires the expression of unique transmembrane receptors and adhesion proteins that mediate axon guidance, dendritic branching, and synapse formation. These key developmental processes will be important to examine going forward, given the postulated contributions of cell adhesion proteins to neurodevelopmental and neuropsychiatric disorders [47,48,49].

## 3. Striatal Cholinergic Interneuron Anatomy

In the late 1800s, Kölliker described the presence of sparse “giant interneurons” in the striatum based on Golgi impregnation studies [50]. It was not until much later that these giant interneurons were associated with their neurotransmitter profile, due to difficulties with targeting and anatomically visualizing cholinergic structures. After ChAT was successfully purified from the brain and antibodies were generated against it [51], the “giant interneurons” of the striatum were confirmed to be cholinergic in rodents, cats, and macaques using immunohistochemistry [52,53,54,55].

ChINs, although being quite morphologically heterogeneous, are commonly referred to as “large aspiny neurons” due to their large somata (~25–40 µm) [56,57] and relatively smooth dendrites [55,58]. ChINs typically have three to five primary neurites [58] and extensive arbors extending hundreds of microns away from the soma [58,59] (Figure 1a–c). Despite their “aspiny” nickname, ChINs possess sparse long thin spines and spine-like filopodia [58,60,61] (Figure 1d,e). Electron microscopy has shown that cortical and thalamic afferent terminals closely appose ChINs [59,62]. Corticostriatal terminals express vesicular glutamate transporter 1 (Vglut1), whereas thalamostriatal terminals express vesicular glutamate transporter 2 (Vglut2). This complementary expression of transporter proteins makes it possible to examine the anatomical distribution of cortical versus thalamic inputs, and also to genetically target/manipulate these inputs separately. ChINs are generally located within the striatal matrix or at patch-matrix borders, with some distal dendrites occasionally terminating within patches (i.e., striosomes) [57]. Likewise, Vglut1- and Vglut2-postive terminals (many of which target ChINs) are more numerous within the striatal matrix [63].

When it comes to neurotransmitter release, ChINs are estimated to have 500,000 axonal varicosities each [13,55]. ChINs have been described as having a “spidery” appearance due to their incredibly fine axonal arborizations that can be hard to fully visualize using standard biocytin filling methods [60,64]. ChINs do not make many conventional synapses and they are believed to participate mostly in volume transmission via ACh diffusion [65]. However, ACh can act in a fast, synaptic-like fashion at dopamine (DA) containing axons via nicotinic ACh receptors (nAChRs) [66,67]. ChINs have also been shown to co-release GABA and glutamate [68]. Glutamatergic co-release is mediated by expression of vesicular glutamate transporter 3 (*Vglut3*), which is rarely expressed in glutamatergic neurons [68]. VGLUT3 colocalizes with the vesicular acetylcholine transporter (VAChT) at presynaptic sites, suggesting that they are acting synergistically [69]. As mentioned earlier, a recent study suggested that 50% of ChINs express *ChAT*, *GAD65*, *Lhx6*, and *Lhx8* and are labeled by antibodies against GAD and VGAT. These ChINs have been termed “dual transmitter cholinergic and GABAergic interneurons” [44].

## 4. Electrophysiological Properties of Striatal Cholinergic Interneurons

### 4.1. Unique Electrophysiological Profile

ChINs have a very well-characterized electrophysiological signature as they have been extensively studied in vitro and in vivo. ChINs have a depolarized resting membrane potential (−55 to −60 mV) and are tonically active [60]. They can fire spontaneous action potentials (APs) in the absence of synaptic input [64] and are sometimes referred to as “autonomous pacemakers” because of this intrinsic AP firing capability [70]. Autonomous pacemaker cells express a particular combination of voltage-gated (Kv, Nav, Cav) and Ca^2+^-activated ion channels that perpetuate rhythmic AP discharge. ChINs have prominent hyperpolarization-activated cyclic nucleotide-gated (HCN) currents [57,64,71] and exhibit large, slow afterhyperpolarization potentials (AHPs) which contribute to their pacemaking activity [10,57]. Here, we briefly summarize some of the key ion channels that ChINs express, as perturbations to the expression or conductance of these channels have been associated with pathophysiological mechanisms in models of striatal dysfunction.

The key conductances that shape ChIN firing can be discussed in the order of AP waveform events. First, the depolarization that precedes AP initiation is thought to be mediated by HCN, Kv4, and Nav channels. HCN channels open in response to the deep hyperpolarization that occurs following an AP and mediate a cation influx (predominantly Na^+^) that depolarizes the membrane potential towards the AP threshold. Neurons with dominant HCN currents often fire APs at a high frequency, as HCN channels allow for rapid repolarization between successive APs. In line with this, blocking HCN channels in ChINs with the compound ZD-7288 causes an increase in the duration of AHPs, i.e., a decrease in the AP firing frequency [72]. Despite the presence of HCN channels, ChINs have a relatively slow AP discharge frequency (3–10 Hz) due to the presence of A-type K^+^ currents mediated by Kv4 channels [9,60,73]. When Kv4 channels open, the resulting K^+^ efflux slows down HCN-mediated depolarization.

Once the membrane potential eventually reaches a certain level of depolarization, Nav channels open and trigger the upstroke of the AP. The AP itself is also shaped by Cav influx and associated Ca^2+^-activated K^+^ conductances via small conductance Ca^2+^-activated K^+^ (SK) channels and big conductance Ca^2+^-activated K^+^ (BK) channels. BK channels are associated with ChIN AP repolarization, and SK channels generate medium AHPs (mAHPs) that help to regulate the firing rate of the neuron [74]. Surprisingly, hyperpolarization-activated inwardly rectifying K^+^ (KIR) currents serve to regeneratively hyperpolarize the cell via their outward currents, amplifying the AHP and deepening its hyperpolarization to the point that HCN channels open and start to depolarize the membrane potential towards threshold again, perpetuating the rhythmic AP firing.

### 4.2. Cholinergic Interneuron “Pauses”

The behavioral correlate of ChIN electrophysiology is the “pause” response. ChINs are known to pause their firing in response to salient sensory cues predictive of reward or aversion during goal-directed learning tasks [9,75,76]. This pause is thought to encode the saliency of sensory stimuli, modulate plasticity at corticostriatal synapses, and modify striatal output to the basal ganglia. All these processes are critical for goal-directed learning, cognitive flexibility, and habit formation. Thus, due to the unique coupling between ChIN AP discharge and motivationally relevant events, the ChIN pause has generated a great deal of interest and has been studied extensively in vivo and in vitro.

ChIN pauses are heterogeneous and it is important to note that they can also present as a transient reduction in AP discharge, and not necessarily a complete cessation of firing [77]. Pauses can be preceded by a “burst” of APs and/or followed by transient “rebound” increases in firing [59,76]. In addition, after repeated learning and conditioning, populations of ChINs over a large striatal area will exhibit synchronized pausing in response to the presentation of the conditioned sensory cue [76]. The heterogeneity and synchronicity of ChIN/TAN pauses depend on many factors, including the nature of the stimulus and reward, its associative context, the task itself, and the degree of conditioning. For example, if a cue associated with an appetitive outcome is swapped for an aversive outcome, the resulting pause duration is shorter on average and has a shorter latency [77,78]. Notably, “longer” pauses in response to rewarding outcomes and “shorter” pauses in response to aversive outcomes can both be observed in the same ChIN/TAN, suggesting that pauses encode context-relevant information and are not purely mediated by intrinsic conductances.

There have been many hypotheses about precisely which factors mediate and modulate pauses in firing, which is known to be dependent on intact dopaminergic [75] and intralaminar thalamic innervation [79]. For example, pausing behavior can be triggered by synaptic inputs and is modulated by (DA) [80]. In monkeys, the pause in ChIN/TAN firing is associated with phasic activity in DA neurons [78,81], and the stimulation of DA release in acute brain slices can evoke ChIN pauses via D2 receptors [82,83]. It has also been suggested that GABAergic inputs (to accumbal ChINs) [84] and/or excitatory glutamatergic inputs [85,86] are sufficient to trigger pauses. Thus, the ChIN/TAN pause response is most likely a product of the complex interplay between corticostriatal and/or thalamostriatal inputs, inhibitory inputs, and intrinsic conductances that are highly sensitive to DA modulation [87].

It was also recently shown that ChINs pause in response to transient shifts in excitatory and/or inhibitory synaptic tone, where withdrawal of excitatory synaptic input is sufficient to induce a pause [88]. This is not mediated by currents that drive AHPs following single APs, but rather by a slow, delayed rectifying, outward potassium (K^+^) current (I_Kr_) carried by Kv7.2 or Kv7.3 channels [88]. These findings also provide a model for DA modulation of pausing behavior, whereby DA is more likely to augment pauses by acting presynaptically to strengthen excitatory inputs. This potentiation of excitatory synaptic strength can promote K^+^ efflux (I_Kr_) when the excitatory tone is transiently pulled back [88], and this K^+^ efflux is strong enough to hold off APs against the weakly depolarizing Na^+^ current [72]. In summary, the complex mechanisms that shape ChIN pauses are still an area of significant interest, as they provide a neurobiological readout of learning behavior. Alterations to synaptic and intrinsic processes surrounding ChIN pauses are reported in several disorders, which will be discussed further in Section 7.

## 5. Synaptic Inputs and Neuromodulation

Cholinergic modulation of striatal circuitry is governed by the intrinsic properties of striatal ChINs, and by the excitatory, inhibitory, and neuromodulatory inputs that they receive. ChINs and MSNs receive inputs from similar extrastriatal regions, but compared to MSNs, ChINs receive a greater amount of intrastriatal inputs [89]. ChINs are able to modulate MSN activity directly via muscarinic ACh receptors (mAChRs) [17,90,91], and presynaptically via mAChRs and nAChRs on glutamatergic terminals that synapse onto MSNs [17,66,67,92,93,94]. The dynamic balance of excitatory and inhibitory synaptic tone onto ChINs is a large determinant of their excitability and impact on striatal output.

### 5.1. Excitatory Connections

ChINs receive excitatory synaptic input mainly from associative regions of the cortex and intralaminar thalamic nuclei [60,89,95], but also from the pedunculopontine nucleus (PPN) of the brainstem [95] (Figure 2).

There has been some debate about the relative weighting of cortical and thalamic inputs onto ChINs. While most assert that thalamic inputs dominate [62,85,89,96,97], a recent viral tracing study suggests that ChINs and PVINs in the dorsal striatum receive the vast majority of their inputs from cortical neurons [95]. Another study, however, demonstrated that while some individual ChINs do not exhibit robust responses to corticostriatal stimulation, cortical inputs to ChINs have significant effects on striatal output due to the convergence of multiple ChINs onto a single MSN [91]. Corticostriatal and thalamostriatal inputs onto ChINs can be segregated anatomically and electrophysiologically anatomically via their expression of VGlut1 and VGlut2 [98,99], respectively, and electrophysiologically by their neurotransmitter release dynamics. Corticostriatal terminals have a high neurotransmitter release probability and exhibit short term depression with repetitive stimulation. By contrast, thalamostriatal terminals apposing ChINs have a low neurotransmitter release probability and exhibit short-term facilitation with repetitive stimulation [59].

#### 5.1.1. Cortical Inputs

ChINs receive direct corticostriatal inputs, mostly on the spine-like appendages along their distal dendrites [100]. Juxtacellular recordings show that stimulation of cortical afferents can elicit the characteristic burst–pause–rebound spiking activity in ChINs [59]. While ChINs are thought to receive inputs from an array of cortical areas [59,89,95,96], we will focus on inputs from the orbital prefrontal cortex (oPFC) and the anterior cingulate cortex.

The oPFC makes direct glutamatergic synapses with ChINs located in the caudate/putamen and the nucleus accumbens in non-human primates and rats [101], as well as mice [95]. In rats, the oPFC topographically projects to the more ventral and medial parts of the striatum [101]. The oPFC, as well as the prelimbic and infralimbic cortical areas, are essential for behavioral flexibility in response to changing environmental contingencies [102,103,104]. The connection between the oPFC and ChINs is particularly interesting within the context of this review, as deficits in frontostriatal circuits are associated with disorders such as obsessive-compulsive disorder [105] and Tourette disorder [106].

ChINs in the dorsomedial striatum (DMS) have been directly implicated in behavioral flexibility and in signaling state changes [107,108,109]. Okada et al. showed that specific ablation of ChINs in the DMS leads to an enhancement of reversal learning and extinction behaviors in rats [110]. This enhancement could be prevented by infusing a non-specific mAChR antagonist into the DMS at any time during the learning, reversal, or extinction phase of the experiment [110]. Interestingly, another study in rats showed that individual ChINs in the DMS encoded the current state within each behavioral trial [111]. Importantly, intact oPFC connections were required for ChINs to encode environmental state under these conditions. These findings suggest the oPFC and ChINs of the DMS are critically important for recalling associative information about the current state to inform action selection [111].

The cingulate cortex is thought to play a role in signalling when an error has been made or when a specific action fails to produce the desired outcome [112]. The anterior cingulate is rarely activated in isolation, but instead its activity aligns with other areas such as the oPFC and striatum, which together are thought to allow for complex evaluation of outcomes and dynamic decision making processes [112]. Neuroanatomical tracing studies have demonstrated direct connections between neurons of the cingulate cortex and striatal ChINs [89,95]. It was shown by Melendez-Zaidi et al. (2019) that SSTINs also receive inputs from the cingulate cortex. Optogenetic activation of the cingulate cortex terminals drove a pause in SSTIN firing that was dependent on ACh release from ChINs [113]. Therefore, excitatory inputs from the cingulate cortex onto ChINs can pattern SSTIN activity as well, and this can lead to NO release and the induction of long-term depression (LTD) at corticostriatal synapses [114,115]. As such, it begs the question, “when an action does not result in the desired outcome, does activation of the cingulate cortex recruit ChINs and SSTINs to “delete” this striatal “memory?”

#### 5.1.2. Thalamic Inputs

Probably the most robust excitatory input onto ChINs comes from the parafascicular nucleus (PFn), part of the thalamic intralaminar complex [89,95]. The PFn relays sensory information about behaviorally relevant events and, by extension, it also plays a key role in circuits that mediate behavioral flexibility and set-shifting during goal-directed learning [116]. For example, consider an animal that has learned to associate a reward outcome with certain specific conditions. When the prediction error increases, the circuit must promote learning in order to update the accuracy of award expectation for when the same parameters are presented again [117]. PFn inputs onto striatal ChINs are very important for this process [116]. An in vivo microdialysis study showed that inactivation of the PFn projection to the striatum can impair reversal learning, as it blocked the striatal ACh release that would normally occur during this process [118]. In addition, pharmacological inhibition of the PFn abolishes the expression of pausing and rebound spiking activity in ChINs [79], demonstrating that excitatory PFn inputs are likely critical for ChIN pausing in vivo in response to salient sensory stimuli.

A recent study by Tanimura et al. (2019) showed that there are two populations of PFn neurons that differentially target ChINs versus MSNs [119]. PFn inputs can recruit ChINs to potentiate glutamate release at other PFn terminals onto iMSNS via the activation of nAChRs [119]. Under normal dopaminergic conditions, this selective enhancement of excitatory PFn input onto the iMSN is suppressed via D2 receptors on ChINs. This mechanism will be discussed further in Section 7.5.

#### 5.1.3. Brainstem Inputs

The PPN also projects to the striatum [120], but until recently, it was not known which cell type these terminals targeted. The PPN is in the upper brainstem, and has been implicated in a wide range of behaviors including motivation, learning, and movement [121]. Anatomically, it appears that some of these PPN terminals target the somata and proximal dendrites of ChINs [95]. It has been reported that optogenetic stimulation of PPN terminals drives monosynaptic excitatory currents in the majority of ChINs [95].

### 5.2. Inhibitory Connections

While a great deal of focus has been placed on understanding the relative contributions of excitatory synaptic transmission onto ChINs, it is estimated that ~60% of synapses onto ChINs are GABAergic [122]. Striatal ChINs express GABA_A_ receptors [123] and local striatal stimulation produces monosynaptic inhibitory postsynaptic currents (IPSCs) in ChINs that can be blocked with bicuculline, confirming intra-striatal GABAergic synaptic inputs [124]. It is thought that GABA exerts inhibitory effects on ChINs both directly and indirectly; directly via GABA_A_ receptors on the ChIN itself, and indirectly through presynaptic inhibition of corticostriatal terminals via GABA_B_ receptors [123]. Interestingly, ChINs undergo long-term potentiation (LTP) in response to high frequency intrastriatal stimulation that is dependent on D5 DA receptor activation [125], and it has both an excitatory postsynaptic current (EPSC) and an IPSC component [126].

The identities of GABAergic neurons synapsing onto ChINs were only recently revealed. Extrastriatal inhibitory input from the external globus pallidus (GPe) has now been characterized and functionally confirmed via optogenetic activation of GPe neurons, which elicited monosynaptic IPSCs in all recorded ChINs [95]. In addition, optogenetic stimulation of GPe terminals was sufficient to reduce tonic firing and trigger rebound firing in ChINs [95]. Local striatal inputs to ChINs include MSNs [19,122,127], THINs [128], and NPY-NGFINs [92,129] (Figure 2). PVINs rarely make efferent connections with ChINs. The divergent connectivity from the reticular thalamus and PFn onto PVINs and ChINs, respectively, situates these striatal interneurons in parallel functional circuits [95].

ChINs have close functional associations with striatal interneurons and are embedded in networks of cholinergic and tyrosine hydroxylase (TH)-positive fibers [130]. Recurrent inhibitory networks between ChINs and THINs can drive polysynaptic GABAergic inhibition [131], as a single ChIN spike is sufficient to drive inhibition in neighboring ChINs via THINs [128]. More recently, ChIN–THIN–ChIN disynaptic transmission has been shown to induce synchronized pauses in firing across multiple ChINs which, interestingly, can be attenuated by a D2-mediated reduction in ACh release [128]. Finally, optogenetic stimulation of SSTINs (also known as low-threshold spiking interneurons) reliably trigger IPSCs in ChINs [129].

Optogenetic activation of MSNs also leads to postsynaptic IPSCs in the majority of ChINs [127], suggesting that axon collaterals from MSNs frequently synapse onto these neurons. It has been reported that substance P-positive terminals are more prevalent than enkephalin-positive terminals, which suggests that ChINs receive a heavier weighting of D1 (direct pathway; dMSN) versus D2 (indirect pathway; iMSN) input [132]. However, the direct and indirect pathways are not as discrete and opposing as once thought [133,134].

### 5.3. Neuromodulation

ChINs are modulated by a whole suite of neuroactive compounds including (but not limited to) DA, glutamate, substance P, enkephalin, dynorphin, serotonin, noradrenaline, and nitric oxide [135]. Here, we will briefly review some postulated neuromodulatory mechanisms. A recent study by Francis et al. (2019) elegantly dissected a circuit in the nucleus accumbens (NAc) whereby D1-MSNs can induce LTP in D2-MSNs via the intermediary activation of ChINs [136]. This was reported to occur via substance P release from D1-MSNs, which binds to neurokinin 1 receptors on ChINs, inducing an increase in ChIN firing. This subsequent enhancement of ACh release was sufficient to induce LTP in D2-MSNs through activation of M1 mAChRs. This mechanism was found to be highly specific to the NAc core—it did not even apply to ChINs in the NAc shell.

Dopaminergic terminals in the striatum also have regionally specific effects on ChINs [137]. In the DMS, DA induces pauses in autonomous ChIN firing through activation of D2 receptors, whereas in the DLS, co-release of glutamate from dopaminergic terminals induces burst firing in ChINs via the metabotropic glutamate receptor, mGluR1 [90,138].

## 6. Synaptic Outputs and Neuromodulation

The PPN and the laterodorsal tegmental nuclei send cholinergic projections to the striatum [120,139]; however, ChINs are still the primary source of striatal ACh. Almost every neuronal element in the striatum (including axon terminals of afferent neurons projecting to the striatum and ChINs themselves) expresses nAChRs and/or mAChRs [17]. Activation of nAChRs permits rapid (in the order of milliseconds) synaptic transmission, and mAChRs allow for slow (in the order of seconds) neuromodulation via metabotropic signaling pathways [140]. ACh plays a major role in modulating local striatal circuits via many mechanisms including excitability changes, promoting or inhibiting neurotransmitter release at terminals, and mediating synaptic plasticity. To add to the complexity, ChINs can also co-release GABA [141] and glutamate [68,142], in addition to ACh. ChINs are involved many different modes of synaptic plasticity in the striatum. Altered striatal synaptic plasticity has been observed in hyperkinetic disorders such as Huntington’s disease, dystonia, and levodopa-induced dyskinesias in Parkinson’s disease [143]. By extension, it is thought that hyperkinetic disorders may result from the lack of bidirectional synaptic plasticity and the pathological strengthening of excitatory corticostriatal synapses [144]. The modulatory action of ChINs at corticostriatal terminals is critical for balancing excitation within the striatum.

Extensive work has been done on the cholinergic modulation of striatal output, although it has been difficult to comprehensively describe the output connections of ChINs within the striatum because they do not tend to form traditional synaptic contacts with other neurons. However, based on paired recordings and optogenetic studies, we now have a functional connectivity map that includes ChINs, MSNs, NPYINs, PVINs, SSTINs, as well as midbrain dopaminergic, corticostriatal, and intralaminar thalamic terminals. Here, we will summarize major and recent findings that pertain to our later discussion of cholinergic pathophysiology in the striatum. See Figure 3 for a simplified diagram showing principal output connections of striatal ChINs.

### 6.1. The Medium Spiny Neuron

It has been recently suggested, based on in vivo Ca^2+^ imaging data, that ChINs are important for synchronizing MSN activity that suppresses or ends a movement bout [145]. There is a plethora of mechanisms by which ChINs can influence MSN activity that, when discussed in parallel, they may seem counteractive. However, it is important to account for the regional, temporal, and contextual specificity of activity within local striatal circuits.

Broadly, ChINs can: (a) modulate MSN activity directly via M1 and M4 mAChRs [17,90,94]; (b) inhibit MSNs by recruiting intermediary GABAergic interneurons [92,146]; (c) induce LTP in MSNs via M1 receptor signaling [147]; (d) inhibit LTP in dMSNs via M4 signaling [148]; (e) promote LTD at MSNs via endocannabinoid signaling [149,150,151]; (f) inhibit glutamate release from presynaptic terminals inputting onto MSNs via M2 mAChRs [93,152,153,154]; (g) gate DA release to modulate MSNs via D1 and D2 receptor signaling [66,140].

It has been shown that multiple ChINs converge on a single MSN [91], which allows concerted ChIN activity to heavily influence striatal output. MSNs of both the direct (dMSN) and indirect (iMSN) pathways express M1 and M4 mAChRs, but dMSNs are more likely to express M4 mAChRs [4,155,156]. Activation of M1 receptors on dMSNs and iMSNs has an excitatory effect, whereas activation of M4 receptors on dMSNs is generally inhibitory [17]. M1 signaling results in a concerted modulation of dendritic and somatic conductances that increase membrane impedance and shifts the resting membrane potential towards the spike threshold. This primes the MSN to spike in response to incoming excitatory inputs that might normally be shunted in a state of lower membrane impedance [17]. M4 mAChRs are abundant in the striatum and are densely packed around the axospinous glutamatergic synapses of MSNs [155,157]. Activation of M4 receptors on dMSNs leads to LTD induction and the suppression of LTP [148].

Based on work in normophysiological and Parkinson’s disease models, it appears that ChINs co-express A2A adenosine receptors and D2 DA receptors. Combined agonism at D2 receptors and antagonism at A2A receptors can converge to reduce ChIN firing in vitro and, presumably, ACh release [158]. The reduction in ChIN firing may serve to withdraw M1 mAChR-mediated inhibition of MSNs, which leads to a transient rise in intracellular Ca^2+^. If prolonged, this rise in Ca^2+^ may be sufficient to drive endocannabinoid release from MSNs, which acts to presynaptically inhibit glutamate release at corticostriatal/thalamostriatal terminals [158]. This mechanism requiring the modulation of ChINs emphasizes their critical role in influencing synaptic plasticity in the striatum, and how alterations in plasticity may contribute to various neuropathologies and be targeted in future therapeutic strategies.

It is well accepted that ACh and DA often work in concert to modulate spike timing, synaptic plasticity, and striatal output. As such, the cholinergic modulation of MSNs depends greatly upon endogenous DA. Recent work by Cai and Ford (2018) showed that dopaminergic modulation of ChINs differentially affects dMSNs of the dorsolateral (DLS) and the dorsomedial striatum (DMS) [90]. DA terminals in the DMS primarily inhibited ChINs via D2 receptors, which impacted their M4-mediated modulation of connected dMSNs. By contrast, DA terminals in the DLS had an excitatory influence on ChINs, via the co-release of glutamate which acted on mGluRs [90]. This regional difference in dopaminergic modulation of cholinergic transmission is especially important to consider for behaviors that recruit regionally distinct striatal subcircuits.

Corticostriatal and thalamostriatal terminals onto MSNs express M2 mAChRs [17]. M2 receptor activation at these terminals in response to a single ChIN spike reduces glutamate release onto the MSN [154]. Due to the autonomous pacemaking activity of ChINs in vivo, there is tonic activity at these M2 receptors and likely inhibition at glutamatergic synapses onto MSNs [154]. The application of M2 mAChR antagonists or suppression of ChIN activity both lead to increased excitatory activity in MSNs [17], though suppression of ChIN activity has multiple downstream effects, as we will discuss. ChIN firing is sensitive to salient cues and changes in environmental state and is encoded in the form of pauses and bursts mediated by dopaminergic and intralaminar inputs, respectively. By extension, this highly patterned ChIN activity provides an elegant means of orchestrating complex presynaptic modulation at MSNs.

### 6.2. Local Inhibitory Interneurons

ChINs have been shown to make nicotinic synapses with PVINs, NPY-NGFINs [92], THINs, and a novel type of inhibitory interneuron subtype labelled with either 5HT3Ra-Cre or 5HT3a-Cre, known as fast-adapting interneurons (FAIs) [159,160,161]. ChINs recruit a number of local inhibitory interneurons via their nAChRs to drive GABAergic inhibition of MSNs or other ChINs. It had been proposed that PVINs were a good candidate cell type for mediating the GABAergic inhibition between ChINs and MSNs [17]. Indeed, ChINs form synapses onto PVINs [162] and ACh excites PVINs via nAChR activation [163]. Optogenetic stimulation of striatal ChINs evokes polysynaptic IPSCs in MSNs [92]. On careful inspection, these IPSCs consisted of a fast and a slow component, which suggested that two mechanisms of inhibition were summating [92]. The slow IPSC component was found to be mediated by NPY-NGFINs [92], while the fast IPSC component was later shown to be mediated by FAIs [159]. Recent work by Dorst et al. (2020) has shown that ChINs form a recurrent inhibitory network with neighboring ChINs by recruiting THINs [128], as discussed above. Interestingly, this polysynaptic connection can be silenced presynaptically by the action of DA on ChIN D2 receptors [128], as discussed above. This emphasizes the complex balance between ACh and DA and how these two neuroactive compounds can have significant effects on many striatal microcircuits.

### 6.3. The Midbrain Dopaminergic Axon

Synchronous ChIN activity has been shown to drive DA release from midbrain dopaminergic terminals via activation of nAChRs [66]. The ability of ACh to modulate DA release is of considerable interest, and it is thought that ACh can act in highly specific areas of the DA axon field and can elicit DA release independent of the SNc neuron’s activity. By extension, stimulation of corticostriatal and thalamostriatal inputs to ChINs can drive DA release from dopaminergic axons, which underscores the gatekeeping role of ChINs in the striatum [164]. ChINs can also drive co-release of GABA from dopaminergic axons that was thought to inhibit MSNs [165], although more recent work by Faust et al. (2016) suggests that NPY-NGFINs and FAIs are the source of ChIN-mediated GABAergic inhibition onto MSNs [160].

### 6.4. Automodulation

ChINs regulate their own ACh release and influence ACh release from neighboring ChINs via the expression of M2 autoreceptors [153,157,166]. When extracellular ACh is high, M2 signalling acts to inhibit further ACh release via a fast, membrane-delimited pathway [167]. This occurs by the shutdown of Ca^2+^ conductances that usually promote ACh release [163], and also potentially by the inhibition of HCN current which reduces ChIN firing [168].

## 7. Cholinergic Interneuron Dysfunction

Dysfunction of cholinergic signaling in the striatum is associated with a number of neurological disorders that affect movement, learning, cognition, and behavior. It is imperative that we examine the role of ChINs within both normo- and patho-physiological models in order to determine their potential value as therapeutic targets. Striatal ChINs are implicated in Tourette disorder, dystonia, attention-deficit hyperactivity disorder, obsessive-compulsive disorder, autism spectrum disorder, Parkinson’s disease, and Huntington’s disease. Despite their diverse etiologies, striatal cholinergic pathophysiology repeatedly presents in preclinical models. Motor stereotypies, for example, are a common feature of autism spectrum disorder, obsessive-compulsive disorder, Huntington’s disease, and addiction, and have been associated with ChIN activity [169]. Recently, it was shown by Crittenden et al. (2017) in mice that under normal conditions, optogenetic activation of ChINs substantially influences striosomal MSN AP patterning in a manner that preserves behavioral flexibility. This cholinergic control is compromised in animals exhibiting drug-induced motor stereotypies, and this contributes to the impaired ability to bias action selection in response to salient cues [170].

### 7.1. Tourette Disorder (TD)

TD is a prevalent neurodevelopmental disorder, frequently considered to be a neuropsychiatric disorder or a movement disorder depending upon the medical perspective. A hallmark feature of TD is the presence of involuntary motor and vocal tics that are typically precipitated by unpleasant sensations described as “*urges*” or “*inner tension*” that serve to elicit tic behavior [171]. TD is highly heritable and whole-exome sequencing has identified several major-risk genes with multiple de novo variants in affected individuals [172]. It is estimated that 12% of TD cases will carry a rare de novo variant in upwards of ~400 distinct genes [172]. TD is highly comorbid with attention-deficit/hyperactivity disorder (ADHD), obsessive-compulsive disorder (OCD), and mood and anxiety disorders [173], suggesting that these disorders can result from perturbations within shared genetic networks and/or overlapping neural circuits [174].

The pathophysiology of TD is thought to involve cortico-basal ganglia-thalamocortical circuitry [175]. Magnetic resonance imaging studies have reported reductions in caudate nucleus volume in children and adults with TD [176,177], although it is possible that these changes are compensatory in nature. Neuroimaging of TD brains show greater functional interactions between brain regions (more connections, stronger connections), and greater functional disorganization of cortico-basal ganglia networks [178,179]. In addition, the pattern of disorganization correlates with the presence of comorbid disorders, examples of which include orbito-frontal changes in patients with TD and OCD [180].

Tics have been postulated to be aberrant habitual responses to unpleasant sensory phenomena known as premonitory urges. A recent preprint by Scholl et al. (2021) details a computational model that supports increased dopaminergic tone within the striatum for driving the kind of enhanced habit formation thought to occur in TD [181]. This idea is especially interesting given the roles of ChINs for gating DA release and habit flexibility [182], and supports the hypothesis that ChIN dysfunction may contribute to TD.

Multiple neurotransmitter systems have been associated with TD, but the strongest association is with DA [183]. DA modulates ChIN activity primarily through action on D2 receptors [180]. According to the Barbeau model, tics could be an expression resulting from increased striatal DA and decreased striatal ACh [184], similar to other hyperkinetic disorders such as Huntington’s Disease and tardive dyskinesia. Evidence supporting a hyperdopaminergic-hypocholinergic state in TD is varied, but tics are highly sensitive to drugs targeting both of these neurotransmitter systems. Neuroleptic drugs currently used to treat TD include risperidone or haloperidol (Haldol) and these function as D2 receptor antagonists, while mAChR antagonists [180] as well as cholinesterase inhibitors [185] have been shown to affect tic expression as well.

It has been reported that a small cohort of adults with severe TD showed a 50–60% loss of striatal ChINs and PVINs in postmortem brain samples [186]. Interneuron loss was restricted to the associative and sensorimotor striatum, whereas the limbic striatum was unaffected. These findings should be interpreted with caution, as most of these individuals were heavily medicated adults with severe TD. Nonetheless, these findings have been modeled in rodents and non-human primates to various degrees to investigate whether striatal interneuron loss could be a key factor in TD pathophysiology. For example, focal application of bicuculline (a GABA_A_ receptor antagonist), to the dorsal striatum can elicit motor tic-like stereotypies in rodents and non-human primates (NHPs), whereas application to the nucleus accumbens in NHPs can produce vocalizations (i.e., grunting) and network dysrhythmia in cortico-striatal-thalamocortical circuits [187,188].

Targeted ablation of ChINs in the DLS can result in self-directed tic-like stereotypies after acute stress or amphetamine challenge [189], while more complete ablation throughout the DMS and DLS along the anterior–caudal striatal axis results in repetitive and ritualistic social behaviors that are reminiscent of TD with comorbid OCD [190]. Interestingly, the latter study showed that following ChIN ablation, animals exhibited increased functional connectivity between the motor cortex and DLS, suggesting that cortical inputs may disproportionately recruit striatal circuits involved in generating and selecting motor responses. Findings from human functional magnetic resonance imaging (fMRI) studies that measured brain activity during spontaneous or mimicked tics, support the idea that heightened functional coupling between associative and motor cortical regions with the caudate and putamen, respectively, may drive compulsive behavior and tic expression [191]. Another study in mice showed that a conjoint depletion of 50% of PVINs and ChINs in the dorsal striatum drove motor-like stereotypies and perturbations to social interaction, reminiscent of autism-spectrum disorder and TD [192]. Interestingly, these changes were restricted to male mice, which may be a recapitulation of the predominant sex bias of these disorders in humans.

### 7.2. Attention-Deficit Hyperactivity Disorder (ADHD)

ADHD is characterized by lack of impulse control, hyperactivity, motor restlessness, and attention deficits [193]. While ADHD has not been directly associated with striatal ChIN dysfunction, its high comorbidity with TD prompts us to briefly summarize what is currently known about its relation to cholinergic systems. ADHD is complex and heterogenous in its presentation and the neural mechanisms that underpin its pathophysiology are currently not well understood, although it is thought to involve dysfunction within cortico-striatal-thalamo-cortical circuits [194,195]. Neuroimaging studies have reported structural abnormalities such as volume reductions in the caudate, putamen, and accumbens, among other changes [196]. Functionally, there are reports of hypoactivation in a number of key brain regions during response inhibition and attention-based tasks [197,198], and reward-based learning tasks [199].

Risk genes have been associated with ADHD, many of which play roles in key neurodevelopmental processes or are associated with dopaminergic and cholinergic systems [200]. The overall genetic evidence for the involvement of cholinergic signaling pathways in ADHD, however, is still weak. Early studies suggested gene variants in the α4 nAChR subunit (CHRNA4) were associated with ADHD [201,202]. In addition, gene variants in SLC5A7, which encodes the choline transporter (ChT) that provides substrate choline for ACh synthesis, have also been suggested to increase the risk for ADHD [203]. Interestingly, ChT is trafficked to the cell membrane at higher rates in the rat cortex during tasks requiring attention [204]. Although these studies suggest associations between genes belonging to cholinergic signaling pathways and ADHD, the findings have not been replicated in larger, population-based genome-wide association studies (GWAS). Nonetheless, nicotine is thought to enhance selective attention though its action on nAChRs [205,206], and mice with a deletion of the α7 nAChR subunit show cognitive deficits and impairments in sustained attention [207,208]. Additionally, a more recent GWAS study found evidence for recurrent variants in the gene encoding for the α7 nAChR subunit, CHRNA7, in association with ADHD [209]. Taken together, these studies provide some evidence for associations between central cholinergic systems, postsynaptic ACh receptors, and choline reuptake transporters with ADHD, but direct evidence for ChIN dysfunction in animal models for ADHD has not been shown.

### 7.3. Eating Disorders

Disordered eating is complex, as it involves both reward and metabolic pathways. Eating disorders can also be characterized in terms of disordered habit formation and behavioral flexibility, two of the key behaviors mediated by striatal ChINs. A recent study implicating the potential role of striatal ChINs in maladaptive eating did so by generating a conditional knock out (cKO) of the vesicular ACh transporter (VAChT) in the dorsal striatum [210]. These animals had impaired behavioral flexibility, enhanced habit formation, and greater vulnerability to maladaptive eating. These changes could be attenuated by administration of donepezil (AChE inhibitor) or L-3,4-dihydroxyphenylalanine (L-DOPA) [210], once again underscoring the importance of DA/ACh interplay in the striatum under both normal and neuropathological conditions. Going forward, it will be interesting to expand on cholinergic mechanisms in models of eating disorders, as these may also apply to neurodevelopmental disorders such as TD.

### 7.4. Obsessive-Compulsive Disorder (OCD)

OCD is a very debilitating neuropsychiatric disorder that is thought to involve orbito-frontal-striatal circuits [101,211,212]. As mentioned above, a recent study that selectively ablated striatal ChINs observed an increase in compulsive behaviors reminiscent of both OCD and TD, paired with an increase in functional connectivity between medial prefrontal, motor cortices, and the DLS [190]. Recent findings suggest that antibodies from children with pediatric autoimmune neuropsychiatric disorder associated with streptococcus (PANDA) bind specifically to striatal cholinergic interneurons [213,214] and reduce their activity [214]. Although it is still controversial, PANDA is thought to result from the production of a post-infection autoimmune response that affects the striatum, and this leads to the sudden onset of tics and OCD. At this stage, the mechanisms by which antibodies binding to ChINs alter their activity are not clear; however, it is interesting to consider that these antibodies might functionally silence the activity of ChINs to some extent, resulting in similar compulsive behavioral outcomes as modelled in the aforementioned ChIN ablation study [190].

### 7.5. Parkinson’s Disease (PD)

PD is a neurodegenerative condition that results from the loss of midbrain dopaminergic neurons in the substantia nigra pars compacta (SNc) that innervate striatal motor regions. This causes motor problems including tremor, rigidity, bradykinesia, postural instability, and gait freezing [215]. PD circuitry is typically described as an imbalance in the direct and indirect pathways, where activity in the indirect pathway dominates [216]. This is thought to be due to changes in synaptic inputs onto MSNs following the loss of DA [216,217], however, frontostriatal changes also contribute to the disease [218]. The current gold standard treatment for PD is L-3,4-dihydroxyphenylalanine (L-DOPA) which acts to “replace” striatal DA. While L-DOPA is temporarily very effective at managing motor dysfunction, long-term use results in the remodeling of striatal circuits, reduced therapeutic benefit, and L-DOPA-induced dyskinesias (LIDs). Although the loss of SNc neurons is thought to be the primary physiological insult, it is now clear that there are other concurrent functional perturbations contributing to the early and late stages of the disease [219].

Notably, the loss of nigrostriatal dopaminergic signaling is met with a rise in intra-striatal cholinergic signaling in the late stages of PD [18,220], in line with Barbeau’s classical see-saw model [184]. Prior to L-DOPA, anticholinergic drugs were used to treat PD. Despite the efficacy of these anticholinergic agents for the treatment of some aspects of PD, these compounds have many unpleasant side effects due to their actions outside of the striatum [221], and thus, are not widely used today. Nonetheless, the importance of cholinergic modulation in both PD and LID pathophysiology is now being thoroughly re-examined as a number of recent studies have elucidated several PD mechanisms implicating striatal ChINs (see recent review by Tubert and Murer, 2020) [222].

There are reported changes to ChIN excitability [223,224,225] and ACh release [226] in PD models. There are also reported losses of neurons in the thalamic intralaminar nuclei of human PD patients [227] and in primate models of PD [228]. Changes to PFn inputs onto ChINs are particularly intriguing to examine within the context of PD, given that recent in vivo data show that ChINs signal transitions between movement states [145,229]. Excitatory PFn inputs onto ChINs also mediate pauses in firing in response to salient stimuli and help gate DA release via nAChRs located on nigrostriatal terminals [107], so these synaptic changes may represent a compensatory response to the loss of DA. Accordingly, it has been suggested that degeneration of PFn inputs to the striatum may underlie some of the set-shifting deficits seen in PD patients [230].

While it has been suggested that there is a specific decrease in PFn inputs onto direct pathway MSNs, elegant work by Tanimura et al. (2019) shows that in a 6-hydroxydopamine (6-OHDA) model of PD, ChINs enhance glutamate release from PFn terminals onto indirect pathway MSNs via α6-subunit containing nAChRs [119]. Normally, this enhancement of glutamate release is downregulated by DA. However, in a DA-depleted state, cholinergic enhancement of PFn excitation of iMSNs is significant. Optogenetic or chemogenetic inhibition of ChINs removed this presynaptic facilitation and partially rescued motor impairments [119,223]. In another 6-OHDA study, PFn to ChIN synapses exhibited reduced NMDA to AMPA receptor ratios compared to controls, suggesting synaptic integration had degraded at those connections [231].

In a study that investigated the role of α-synuclein in early disease pathogenesis, it was shown that transgenically overexpressed truncated or virally introduced human α-synuclein drives a specific interaction with GluN2D-containing NMDA receptors in striatal ChINs that blocks LTP [232]. It was suggested that this mechanism may contribute to motor and cognitive impairments in early stage disease. In a model of progressive DA-loss (*Slc6a3*^DTR/+^), ChIN activity and ACh release were reduced [233]. This unexpected change was thought to be due to DA release from residual dopaminergic axons that activated D1-like receptors on ChINs and impacted AP patterning [233]. However, despite the reduction in activity, the authors described a relative dominance of ACh over DA signaling that drove PD-like pathophysiology [233]. These results implore us to address how specific PD models are recapitulating different aspects of the human disorder. A recent study by Cai et al. (2021) investigated changes to glutamate co-transmission from dopaminergic afferents to ChINs in a low-dose 6-OHDA model of PD [234]. Typically, when glutamate is co-released from dopaminergic axons in the DLS, it activates mGluR1 receptors on ChINs and triggers a burst firing response [90]. However, the burst firing response in ChINs following DA terminal stimulation is lost in the low-dose 6-OHDA model of PD. Interestingly, this was shown to result from a reduction in postsynaptic mGluR1 expression on ChINs. When mGluR1 expression was restored, burst firing responses were rescued and improved the motor impairments that mirror early stage PD.

Striatal ChINs have also been implicated in mechanisms surrounding LIDs. Depending on the stimulation paradigm, optogenetic stimulation of ChINs can either suppress or enhance LIDs by eliciting nAChR desensitization or mAChR signaling, respectively [235]. In a 6-OHDA mouse model of PD treated with levodopa, it has been suggested that there is aberrant burst–pause activity in ChINs associated with LID [236].

### 7.6. Dystonia

Dystonia is a debilitating movement disorder of genetic, idiopathic, or traumatic origin that is characterized by involuntary muscle contractions and postural torsion. Dystonia is thought to involve altered activity in basal ganglia-thalamo-cortical and cerebello-thalamo-cortical circuits, striatal cholinergic signaling, and changes to striatal plasticity [237]. It has recently been shown using PET imaging that there is an age-related decrease in VAChT in the caudate and putamen of human early-onset isolated dystonia (DYT1) patients versus controls, and an age-independent relative decrease in VAChT in the cerebellar vermis of DYT1 patients versus controls [238]. While some forms of dystonia respond well to L-DOPA therapy, the majority of cases are treated with anticholinergic agents, despite their frequent side effects [239].

A major area of interest in models of hyperkinetic disorders is perturbations to normal striatal synaptic plasticity [143], and it is clear that striatal ChINs play a key role in these processes [240]. A study in a *Tor1a* mouse model of dystonia demonstrated impaired corticostriatal plasticity at MSNs that could be rescued through application of M1 mAChR-preferring antagonists [241]. As discussed briefly above (Section 6.1. The Medium Spiny Neuron), ChINs co-express A2A adenosine receptors and D2 DA receptors. The parallel antagonism of A2A together with agonism of D2 is associated with a decrease in ChIN firing that can lead to LTD of corticostriatal terminals at MSNs [158]. This synaptic plasticity mechanism is thought to be altered in models of PD, and recently a similar phenomenon was shown in a rodent model of DYT25 dystonia, where changes to dopaminergic and adrenergic signaling and a loss of corticostriatal LTD was observed [242]. Notably, the potential involvement of striatal ChINs, M1 mAChR, and cannabinoid signaling were not explored, despite the additional observation of impaired motor learning.

While *Tor1a* mutant mice do not exhibit torsional movements, a more severe conditional knockout of *Tor1a* expression in the mouse forebrain does [243]. This resulted in twisting movements reminiscent of DYT1 dystonia, and the loss of striatal ChINs [243]. This study suggests that *Tor1a* is required for normal ChIN development and survival and provides further evidence for a pivotal role for ChINs in the pathogenesis of dystonia. There are many different genetic mouse models used for studying dystonia. For a recent review, see Imbriani et al. (2020) [244]. A convincing demonstration of convergence on striatal cholinergic dysfunction in dystonia was provided recently by Eskow Jaunarajs et al. who used three genetic approaches (*Tor1a*, *Thap1*, and *Gnal*) and saw remodeling of D2 DA receptor signaling in striatal ChINs in all models, as a result of overactive mAChR signaling [245]. A paradoxical increase in ChIN excitability was observed, due to the shift in coupling of the D2 receptor pathway from G_i/o_ to β-arrestin [245]. This paradoxical excitation through D2 receptors has been shown to be restored to inhibitory via application of mAChR antagonists [246]. This explains some of the efficacy of antimuscarinic compounds in humans with dystonia and emphasizes the need for the development of more specific compounds to target these receptors or other signaling molecules in this pathway for therapeutic benefit.

### 7.7. Huntington’s Disease (HD)

HD is a heritable, progressive neurodegenerative disorder characterized by involuntary movements, psychiatric disturbances, and cognitive decline [247]. The pathophysiology of HD is marked by the death of striatal MSNs [248,249]. Striatal ChINs are mostly spared from degeneration [250,251], but they still undergo pathophysiological changes that are thought to contribute to the beginning stages of HD [252]. Early evidence for a hypocholinergic state in HD consisted of reduced mAChR density [253,254] and reduced ChAT activity [255] in the basal ganglia of post-mortem brain tissues. Accordingly, in genetic mouse models of HD, ChINs exhibit reduced ACh release and ACh autoregulation [256], reduced mRNA and protein levels for VAChT and ChAT, and reduced AChE activity [257]. Transgenic mouse models for HD such as R6/2, R6/1, and Q175 have rapidly progressed our understanding of neural mechanisms that may underlie HD [258]. However, it is important to keep in mind that many of these mouse strains do not typically exhibit choreic movements, a key feature of the human disease [259].

Changes to synaptic plasticity in ChINs and MSNs have been reported in a genetic mouse model of HD [260]. Normally, LTP is induced via high frequency stimulation, and in MSNs, this potentiation can then be “depotentiated” with low frequency stimulation. In R6/2 mice, however, LTP was abolished in ChINs and depotentiation was no longer possible in MSNs [260]. The loss of depotentiation was also seen in control animals in the presence of mAChR antagonists or if endogenous ACh was disrupted, demonstrating the key role of ChINs in this process [260]. In turn, they showed that the loss of depotentiation in vitro was occurring in parallel with loss of behavioral flexibility [260]. Humans with HD often exhibit behavioral inflexibility, which might be potentially related to cholinergic dysfunction [261,262], so this is a significant step forward in understanding putative pathophysiological mechanisms in HD.

Loss of synaptic inputs onto ChINs has also been observed in HD mice. In a Q140 knock-in mouse model of HD, ChINs exhibited morphological changes and there were fewer PFn to ChIN synapses in the striatum overall [263]. This is consistent with nerve cell loss in the centromedian-parafascicular (CM-Pf) nucleus of human HD patients [264]; however, it remains unclear which populations within the CM-Pf are affected. In a Q175 model of HD, PFn synapses onto ChINs showed reduced short term facilitation and, by contrast, the stimulation of corticostriatal terminals evoked much larger responses in Q175 mice versus controls [265]. This aberrant strengthening of corticostriatal inputs appears to be mediated by postsynaptic upregulation of Na_V_ channels in ChINs. This alteration did not increase basal firing rates as would be predicted, potentially due to a concomitant increase in Kv4 channel conductance [265]. Overall, a shift in the thalamocortical and corticostriatal excitation of ChINs may drive, at least in part, the hyperkinetic phenotype in Q175 HD models that is characteristic of early stage disease.

It has been suggested that inhibitory GABAergic inputs to ChINs are increased in the R6/2 model of HD, as demonstrated by increased spontanous IPSC frequency and evoked IPSC amplitude [266]. A recent study has also shown that conditional ablation of the mouse Huntingtin gene (Htt) in progenitors expressing the markers Gsx2 or Nkx2.1 results in HD-like symptoms, and the majority of striatal ChINs express Nkx2.1 (see Section 2). Both the Gsx2 and the Nkx2.1 ablation strains exhibited patterns of striatal degeneration reminiscent of human HD, which suggests that perturbations to the development, migration, integration, and survival of these progenitors can contribute to HD pathogenesis [267]. Figure 4 highlights some of the key circuits and behaviors associated with the neurological disorders we have discussed, and how these pathways and behaviors may overlap and recurrently implicate striatal ChINs.

## 8. Conclusions

Central cholinergic circuits are thought to modulate some of the key behaviors, such as learning, arousal, and attention, that allow us to thrive in a dynamic environment. While ChINs only make up 1–2% of the striatal neuronal population, ACh plays a pivotal role at the local circuit level, modulating MSN and local interneuron activity and orchestrating synaptic plasticity. The important contribution of ChINs to striatal physiology is underscored by the many basal ganglia disorders (and now rodent models of those disorders), that point to cholinergic interneuron dysfunction. Common themes that emerge in these studies are: (1) the degeneration and/or functional imbalance of major corticostriatal and intralaminar thalamic synaptic inputs onto ChINs; (2) changes to behaviors that are mediated by ChINs and related circuits; (3) perturbations to synaptic plasticity mechanisms that involve cholinergic signaling; (4) changes to ACh synthesis, release, degradation or reuptake; (5) changes to the intrinsic properties of ChINs. Moving forward, it is, therefore, critical to further dissect the array of intricate mechanisms involving striatal ChINs in models of normal and disordered physiology, as we re-evaluate their potential value as therapeutic targets in a range of neurological disorders.

## Figures and Tables

**Figure 1 cells-10-00907-f001:**
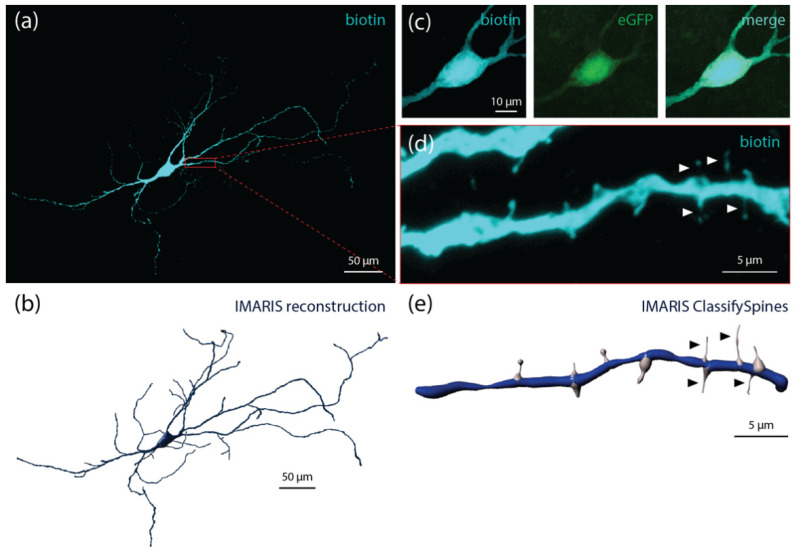
Morphology of a murine striatal cholinergic interneuron. (**a**) Confocal maximum intensity projection of a single biotin-labelled (Biotium) cholinergic interneuron located in the mouse dorsolateral striatum. Biotin was tagged with AlexaFluor™ 555-conjugated streptavidin (Invitrogen) and imaged on a Leica LSM700 confocal microscope. (**b**) Semi-automatic Imaris (Bitplane) 3D reconstruction using *Filaments* and *Surfaces* functions. (**c**) Biotin-labelled neuron (left) expresses enhanced Green Fluorescent protein (eGFP) under control of the choline acetyltransferase (ChAT) promoter (middle). Biotin-AlexaFluor™ signal colocalizes with eGFP signal confirming its cholinergic identity (right). (**d**) Secondary dendrite enlarged from inset shown in (**a**). Spine-like filopodia (arrows). (**e**) Imaris 3D reconstruction of secondary dendrite of interest, where spine-like structures were identified and classified using *ClassifySpines* plugin. Identified spine-like filopodia (arrows).

**Figure 2 cells-10-00907-f002:**
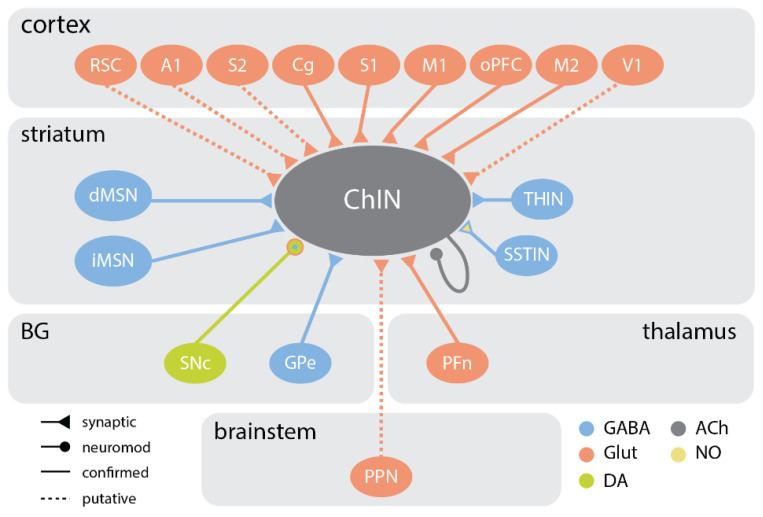
Schematic of major inputs to striatal cholinergic interneurons. Cortical inputs to cholinergic interneurons (ChINs) include the cingulate cortex (Cg), orbital prefrontal cortex (oPFC), primary somatosensory (S1) and motor (M1) cortices, and secondary motor cortex (M2). Putative cortical inputs to ChINs include the secondary somatosensory cortex (S2), primary visual (V1) and auditory (A1) cortices, and the retrosplenial cortex (RSC). Other extrastriatal inputs to ChINs include basal ganglia (BG) structures substantia nigra pars compacta (SNc) and external globus pallidus (GPe), the parafascicular thalamic nucleus (PFn), and the brainstem pedunculopontine nucleus (PPN). Intra-striatal inputs to ChINs include medium spiny neurons (MSN), tyrosine hydroxylase-expressing interneurons (THIN), somatostatin-expressing interneurons (SSTINs; or LTSI), and ChIN automodulation. Note: connections shown based on neuroanatomical tracing and optogenetic mapping studies [89,95].

**Figure 3 cells-10-00907-f003:**
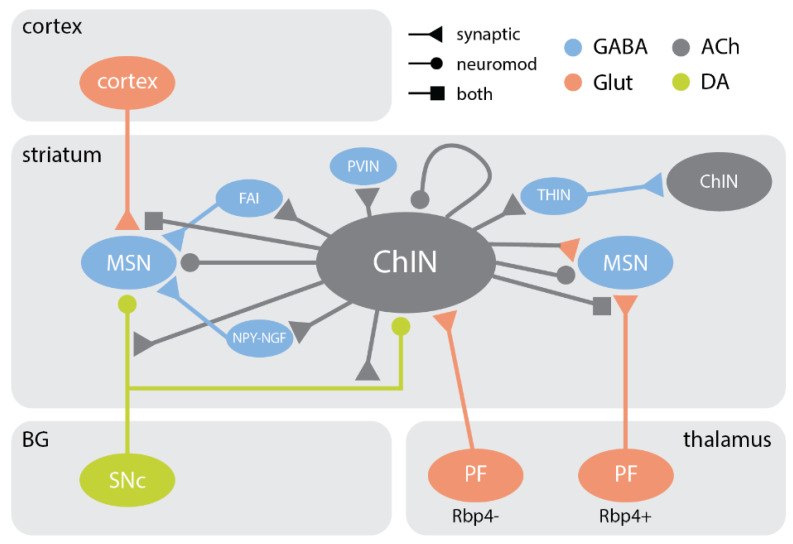
Simplified output map of striatal cholinergic interneurons. Cholinergic interneurons make local connections within the striatum, both conventional synaptic connections that involve nicotinic acetylcholine (ACh) receptors, and volume transmission sites that involve muscarinic ACh receptor signaling. At corticostriatal and thalamostriatal terminals onto MSNs, both nicotinic and muscarinic ACh transmission occur. Abbreviations: MSN = medium spiny neuron, SNc = substantia nigra pars compacta, FAI = fast-adapting interneuron, NPY-NGF = neuropeptide Y-expressing neurogliaform interneuron, PVIN = parvalbumin-expressing interneuron, ChIN = cholinergic interneuron, PF = parafascicular thalamic neuron, THIN = tyrosine hydroxylase-expressing interneuron.

**Figure 4 cells-10-00907-f004:**
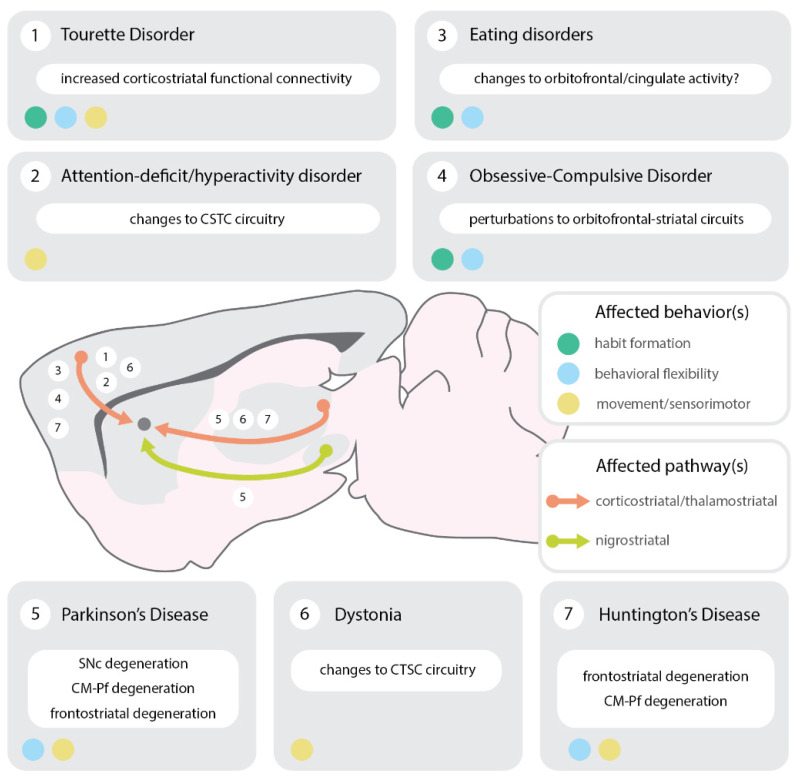
Common circuits and behaviors relevant to striatal cholinergic interneurons are perturbed in common neurodevelopmental, neuropsychological, and neurodegenerative disorders. CSTC = cortico-striatal-thalamo-cortical (circuit); SNc = substantia nigra pars compacta; CM-Pf = centromedian-parafascicular (nucleus); ChIN = cholinergic interneuron. Note: circuits are shown in a mouse brain for illustrative purposes, however, many of these circuits have been shown to be altered in human functional imaging studies.

## Data Availability

No new data were created or analyzed in this study. Data sharing is not applicable to this article.

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
