# Peer review of "Recurrent Implication of Striatal Cholinergic Interneurons in a Range of Neurodevelopmental, Neurodegenerative, and Neuropsychiatric Disorders"

_cells, 2021, doi:10.3390/cells10040907_

Round 1
Reviewer 1 Report
In this review the Authors address the critical role of cholinergic interneurons(ChIs)as key players in the striatal control of attention, goal-directed actions, habit formation, and behavioral flexibility.
The Authors describe alterations of striatal cholinergic interneurons associated with many neurodevelopmental, neurodegenerative, and neuropsychiatric disorders. They report that,although therole of acetylcholine (Ach) has been recognized in many of these disorders,the possible use of drugs targeting Ach has been limited by adverse side effectsof these drugs.Nevertheless, the Authors provide an interesting and accurate update on striatal cholinergic interneuron function, connectivity, and their putative involvement in several neurological disorders. This manuscript is clear and well written. It addresses most of the key aspects dealing with ChIsin a critical manner.
Figures are of good quality. The reference list is generally adequate (but see comments).In conclusion, this is a timely and interesting review that might be interesting for both basic scientists as well as for clinically-oriented neuroscientists. I have only fewsuggestions for the Authors:1)While the
Authors explore most of the neurological and psychiatric disorders possibly associated with dysfunctions of ChIs, they do not address the role of this neuronal subtype in dystonia. In thisneurological dysfunction ChIsplay a critical role and antimuscarinic drugs are still in use as therapeutic agents. Several references might help in addressing this issueand should be quoted:
Mazere et al. Striatal and cerebellar vesicular acetylcholinetransporter expression is disrupted in human DYT1 dystonia. Brain. 2021 Feb26:awaa465. doi: 10.1093/brain/awaa465.
Calabresi P, Standaert DG. Dystonia and levodopa-induced dyskinesias in Parkinson's disease: Is there a connection? Neurobiol Dis. 2019 Dec;132:104579. doi: 10.1016/j.nbd.2019.104579. Epub 2019 Aug 22. PMID: 31445160; PMCID: PMC6834901.
Downs et al. Theneurobiological basis for novel experimental therapeutics in dystonia. NeurobiolDis. 2019 Oct;130:104526. doi: 10.1016/j.nbd.2019.104526. Epub 2019 Jul 4. PMID:31279827; PMCID: PMC6885011.
Calabresi et al. Hyperkinetic disorders and loss of synaptic downscaling. Nat Neurosci. 2016 Jun28;19(7):868-75. doi: 10.1038/nn.4306. PMID: 27351172.
Eskow Jaunarajs. Striatalcholinergic dysfunction as a unifying theme in the pathophysiology of dystonia.Prog Neurobiol. 2015 Apr;127-128:91-107. doi: 10.1016/j.pneurobio.2015.02.002.Epub 2015 Feb 17. PMID: 25697043; PMCID: PMC4420693.
Sciamanna et al. Rhes regulatesdopamine D2 receptor transmission in striatal cholinergic interneurons.Neurobiol Dis. 2015 Jun;78:146-61. doi: 10.1016/j.nbd.2015.03.021. Epub 2015 Mar26. PMID: 25818655.
Maltese et al. Anticholinergic drugs rescue synapticplasticity in DYT1 dystonia: role of M1 muscarinic receptors. Mov Disord. 2014Nov;29(13):1655-65. doi: 10.1002/mds.26009. Epub 2014 Sep 4. PMID: 25195914.2)
Key experimental studies showing the role of ChI in synaptic transmission and plasticity have not be included in the review and should be discussed and quoted:
Bonsi et al. Modulatory action of metabotropic glutamate receptor (mGluR) 5 on mGluR1 function in striatal cholinergic interneurons. Neuropharmacology. 2005;49 Suppl 1:104-13. doi: 10.1016/j.neuropharm.2005.05.012. PMID: 16005029.
Bonsi et al. Coordinate high-frequency pattern of stimulation and calcium levels control the induction of LTP in striatal cholinergic interneurons. Learn Mem. 2004 Nov-Dec;11(6):755-60. doi: 10.1101/lm.82104. Epub 2004 Nov 10. PMID: 15537735; PMCID: PMC534704.
Centonze et al. Receptor subtypes involved in the presynaptic and postsynaptic actions of dopamine on striatal interneurons. J Neurosci. 2003 Jul 16;23(15):6245-54. doi: 10.1523/JNEUROSCI.23-15-06245.2003. PMID: 12867509; PMCID: PMC6740558.
Pisani et al. Activation of beta1-adrenoceptors excites striatal cholinergic interneurons through a cAMP-dependent, protein kinase-independent pathway. J Neurosci. 2003 Jun 15;23(12):5272-82. doi: 10.1523/JNEUROSCI.23-12-05272.2003. PMID: 12832552; PMCID: PMC6741190.
Pisani et al. Metabotropic glutamate 2 receptors modulate synaptic inputs and calcium signals in striatal cholinergic interneurons. J Neurosci. 2002 Jul 15;22(14):6176-85. doi:10.1523/JNEUROSCI.22-14-06176.2002. PMID: 12122076; PMCID: PMC6757931.
Pisani et al. Functional coexpression of excitatory mGluR1 and mGluR5 on striatal cholinergic interneurons. Neuropharmacology. 2001 Mar;40(3):460-3. doi: 10.1016/s0028-3908(00)00184-2. PMID: 11166340.
Centonze et al. Stimulation of nitric oxide-cGMP pathway excites striatal cholinergicinterneurons via protein kinase G activation. J Neurosci. 2001 Feb 15;21(4):1393-400. doi: 10.1523/JNEUROSCI.21-04-01393.2001. PMID: 11160411;PMCID: PMC6762226.
Pisani et al. Activation of D2-like dopamine receptors reduces synaptic inputs to striatal cholinergic interneurons. J Neurosci. 2000 Apr 1;20(7):RC69. doi:10.1523/JNEUROSCI.20-07-j0003.2000. PMID: 10729358; PMCID: PMC6772255.
Calabresi et al. Acetylcholine-mediated modulation of striatal function. Trends Neurosci. 2000 Mar;23(3):120-6. doi: 10.1016/s0166-2236(99)01501-5. PMID: 10675916.
Calabresi et al. Endogenous Ach enhances striatal NMDA-responses via M1-like muscarinic receptors and PKC activation. Eur J Neurosci. 1998 Sep;10(9):2887-95. doi: 10.1111/j.1460-9568.1998.00294.x. PMID: 9758158.
Calabresi et al. Striatal spiny neurons and cholinergic interneurons expressdifferential ionotropic glutamatergic responses and vulnerability: implications for ischemia and Huntington's disease. Ann Neurol. 1998 May;43(5):586-97. doi: 10.1002/ana.410430506. PMID: 9585352.
Calabresi et al. Muscarinic IPSPs in rat striatal cholinergic interneurones. J Physiol. 1998 Jul15;510 ( Pt 2)(Pt 2):421-7. doi: 10.1111/j.1469-7793.1998.421bk.x. PMID:9705993; PMCID: PMC2231043)Some key experimental studies showing the role of ChI in experimental models of PD and other neurological dysfunctionshave not beenincluded in the review and should be discussed and quoted:
Tozziet al. Alpha-SynucleinProduces Early Behavioral Alterations via Striatal Cholinergic SynapticDysfunction by Interacting With GluN2D N-Methyl-D-Aspartate Receptor Subunit.Biol Psychiatry. 2016 Mar 1;79(5):402-414. doi: 10.1016/j.biopsych.2015.08.013.Epub 2015 Aug 20. PMID: 26392130.
Tozzi et al. The distinct role of medium spinyneurons and cholinergic interneurons in the Dâ‚‚/Aâ‚‚A receptor interaction in thestriatum: implications for Parkinson's disease. J Neurosci. 2011 Feb2;31(5):1850-62. doi: 10.1523/JNEUROSCI.4082-10.2011. PMID: 21289195; PMCID:PMC6623738.
Di Filippo et al. Impaired plasticity at specific subset of striatalsynapses in the Ts65Dn mouse model of Down syndrome. Biol Psychiatry. 2010 Apr1;67(7):666-71. doi: 10.1016/j.biopsych.2009.08.018. Epub 2009 Oct 8. PMID:19818432.
Calabresi et al. Post-ischaemic long-term synaptic potentiation in the striatum: a putative mechanismfor cell type-specific vulnerability. Brain. 2002 Apr;125(Pt 4):844-60. doi:10.1093/brain/awf073.
Author Response
We thank the Reviewer for their consideration and subsequent improvement of this review.
We have expanded the text to cover the role of striatal cholinergic interneurons in dystonia [see lines 733-73]. In retrospect we agree that omission of dystonia from out discussion would have been an oversight given the strong body of literature, our mention of other movement disorders, and the current prescribed use of anticholinergic drugs for treating this condition.
We have also re-emphasised the importance of striatal cholinergic interneurons for mediating striatal plasticity [see lines 409-416].
Calabresi et al. (1998) are now cited [see line 515] on first mention of M2-mediated automodulation. Calabresi et al. (1998, 2002) are now cited in Section 7.7. Huntington’s disease [see line 778]. Bonsi et al. (2004) are now cited in Section 5.2. Inhibitory connections [see lines 347-9]. Tozzi et al. (2011) are now cited in Section 6.1. The medium spiny neuron [see lines 451-61]. Tozzi et al. (2016) are now cited in Section 7.5. Parkinson’s disease [see lines 709-13].
Reviewer 2 Report
In this manuscript, Poppi and colleagues review the literature on the role of striatal cholinergic interneurons in neurodevelopmental, neurodegenerative, and neuropsychiatric disorders. The manuscript is well written and organized.
There is one suggestion to improve the manuscript further. Although there is very limited evidence, striatal CHIN may be involved in the autism-spectrum disorder. It would be beneficial for the review to cover this topic.
Two important references are missing:
Crittenden et al 2017 https://pubmed.ncbi.nlm.nih.gov/28377698/
Rapanelli et al 2017 https://pubmed.ncbi.nlm.nih.gov/28347488/
Author Response
We thank the Reviewer for their consideration of our review, and for the encouragement to include references associating ASD and models of ASD with striatal cholinergic interneuron dysfunction. We initially intended to comment on these findings but decided against given that the evidence is currently limited, as the Reviewer points out.
We have now included both of the suggested studies, Crittenden et al. (2017) [see lines 538-43] and Rapanelli et al. (2017) [see lines 602-4].
Round 2
Reviewer 1 Report
The manuscript has been improved. According to the suggestions of the reviewers the Authors have properly modified the manuscript.
Reviewer 2 Report
N/A